# Western Spruce Budworm Effects on Forest Resilience

**DOI:** 10.3390/plants11233266

**Published:** 2022-11-28

**Authors:** Adam D. Polinko, Marguerite A. Rapp, Andrew J. Sánchez Meador, Andrew D. Graves, Daniel E. Ryerson, Kristen M. Waring

**Affiliations:** 1School of Forestry, Northern Arizona University, Flagstaff, AZ 86011, USA; 2Forest Health Protection, USDA Forest Service, Albuquerque, NM 87102, USA

**Keywords:** insect defoliation, tree mortality, resilience, species composition

## Abstract

Western spruce budworm (*Choristoneura freemani* Razowski) is the most destructive defoliator of forests in the western US. Forests in northern New Mexico experienced high levels of WSBW-caused defoliation and subsequent mortality between the 1980s and 2010s. The effects of severe western spruce budworm outbreaks on stand dynamics in the US Southwest are still relatively unknown, but understanding the impacts is important to the management and resilience of these forests. To begin addressing this knowledge gap, we conducted a study along two gradients: an elevational gradient from mixed-conifer to spruce-fir forests and a gradient of WSBW-caused defoliation intensity. We recorded overstory and understory stand conditions (size structure, species composition, damaging agents). Western spruce budworm was the primary damaging agent of host trees in all stands andcaused host tree mortality across all size classes, particularly in spruce-fir stands. Results indicate an unsustainable level of mortality in spruce-fir stands and a transition towards non-host species in mixed-conifer stands. Low levels of regeneration coupled with high overstory mortality rates indicate a potential lack of resilience in spruce-fir stands, whereas resilience to future western spruce budworm defoliation events may have increased in mixed-conifer stands affected by these outbreaks.

## 1. Introduction

Native forest insects drive disturbance patterns and stand dynamics in many forested ecosystems [1,2] and play an important role in nutrient cycling, regulating productivity, and production of snags and downed woody debris [3,4,5]. The influence of insects on forest stand dynamics is unique compared to other disturbances, due to spatially synchronous outbreaks, temporal periodicity and specific host selection [5]. The western spruce budworm (*Choristoneura freemani* Razowski; previously *C. occidentalis* Freeman; hereafter: WSBW) is the most widespread native defoliator of forests in western North America [6].

Defoliation by WSBW can cause individual tree growth loss, top kill and mortality, thereby altering stand structure at larger scales [7,8]. Dense, multi-strata stands composed of shade-tolerant species are most susceptible to severe effects of WSBW defoliation [8,9]. Within these complex forest structures, understory and lower-strata trees are particularly vulnerable to heavy defoliation due to the dispersal of larvae from overstory trees [10]. WSBW defoliation events may have secondary effects on trees and stands, including alteration of potential fire behavior [11,12] and increase susceptibility to outbreaks of other forest insects, such as Douglas-fir beetle (*Dendroctonus pseudotsugae* Hopkins) [13].

Ecological resilience refers generally to the capacity for a system to experience disturbance and return to a similar structure and function following the disturbance [14]. Resistance is a related term but can be specifically defined in forested systems as the effect of structure and composition on disturbance severity [15]. We define resistance here as short-term resilience, thus having characteristics that limit disturbance severity [16]. Resistance and resilience of southwestern US (defined here as Arizona and New Mexico) forests have declined in many locations due to the interactions between past management practices, fire exclusion, and climate change [17]. Native defoliators su_=_ch as WSBW may be contributing to declining resistance and resilience and subsequent future contractions in forest cover or forest type conversions [17], although there has been a lack of research on this topic (see [13] for a review on the interactions between biotic disturbance agents, fuels and fire).

Outbreaks of WSBW may be related to climate conditions that either initiate or sustain an outbreak [18]; however, there is also evidence to suggest that management decisions have altered outbreak intensity and duration by creating larger tracts of dense forest dominated by host species [19,20,21]. Fire suppression, grazing, and selective logging of ponderosa pine (*Pinus ponderosa* var. *brachyptera* (Engelm.) Lemmon) since the late 1800s have resulted in widespread and continuous tracts of susceptible forests of fire-intolerant and shade-tolerant host species [22,23]. This trend occurred in both ponderosa pine and mixed-conifer forests throughout the southwestern US, as demonstrated by reconstructions of historical forest conditions [20]. The cumulative effects of these actions caused changes in the developing stand structure and composition. Without the regulatory presence of fire, species that would otherwise be killed in fire events became prolific in the understory and eventually the overstory. The adjacent, higher elevation spruce-fir (*Picea* spp.–*Abies* spp.) forests have experienced fewer shifts, yet less research has focused on this forest type in the US Southwest [24].

The impacts of WSBW outbreaks on stand structure, species composition and regeneration patterns, and therefore resistance and resilience, in southwestern US forests remains relatively unknown. Outbreak impacts in mixed-conifer forests have been studied in other parts of WSBW’s range (e.g., [9,25,26], and the importance of understanding the impacts of WSBW has been recently noted [27]. Previous research in the US Southwest has focused on dendrochronological reconstructions of outbreak timing and seasonality [20,21,28]. We investigated the influences of WSBW on stand structure and composition across two gradients: an elevational gradient from mixed-conifer to spruce-fir forests and a gradient of WSBW-caused defoliation intensity. Specifically, our objective was to determine how the recent WSBW outbreak has affected overall tree mortality, species composition and regeneration. We further assessed resistance and resilience to WSBW in these stands and discuss long-term implications for spruce-fir and mixed-conifer forest types given our findings.

## 2. Results

### 2.1. Stand Conditions, WSBW Defoliation and Other Biotic Damaging Agents

In the sampled stands, host trees comprised between 55–93% of total stand basal area, with a mean of 71% in the mixed-conifer forest type and 78% in the spruce-fir forest type (Table 1). Stand canopy cover ranged from 33.4% to 57.6% with a mean of 47%. Spruce-fir stands had a higher mean canopy cover (49%) than mixed-conifer stands (44%). Cumulative years of detected WSBW defoliation ranged from 7.1 to 14.3 years (Table 1) and were negatively correlated with elevation (Pearson’s correlation = −0.227, *p* < 0.05). Sketch method cumulative defoliation generally increased with cumulative years of detected defoliation (Pearson’s correlation = 0.279, *p* < 0.05). Sketch method cumulative defoliation and 6-class cumulative defoliation were related (Pearson’s correlation = 0.582, *p* < 0.05) but sketch method cumulative defoliation was consistently lower than the 6-class method (Table 2). Only 3.8% of all live overstory trees had cumulative defoliation greater than 50%. Cumulative defoliation was relatively similar throughout all diameter classes (mean 8.2–18.0%) with the lowest defoliation occurring in the largest two diameter classes (mean 8.2% and 9.2%, respectively). Cumulative defoliation was similar among species with no apparent trend (mean 15.7–18.7%). Evidence of bark beetles was highest in corkbark fir (17.8%) and lowest in Douglas-fir (3.8%). Signs and/or symptoms of root disease or other pathogens were highest in corkbark fir (23.8%) and lowest in Douglas-fir (4.3%). Bark beetle or pathogen evidence was found at various levels across most diameter classes but was most consistently found in the lower diameter classes (bar chart, Figure 1). Both bark beetle and pathogen signs and/or symptoms were found in only 3.4% of all living host species, whereas WSBW defoliation was found in 91.9% of all trees (waffle plots, Figure 1).

### 2.2. Overstory Mortality

Mortality was highest in corkbark fir (83.5%) and Engelmann spruce (42.1%) in the higher elevation spruce-fir stands (Table 3). White fir mortality was also relatively high (31.2%), a species found almost exclusively in the mixed-conifer stands (Table 3). Mortality across all species occurred most frequently in the lower diameter classes (Figure 2). The two most shade tolerant species, corkbark fir and white fir, decreased in importance in all stands except for Jose Maria. Nonhost species (quaking aspen and ponderosa pine) generally increased in importance (Figure 3). A multivariate model of live species composition (as defined by live importance value) explained 33% of variation across all forest types. Live species composition was driven by elevation (32%, *p* < 0.05) and 45° transformed aspect (1%, *p* < 0.05). Multivariate models of reconstructed species composition (as defined by reconstructed importance values) explained 39% of variation. Elevation and 45° transformed aspect explained 37% and 1% of variation, respectively (*p* < 0.05). Cumulative years of detected defoliation explained a small but significant amount of variation (1%; *p* < 0.05). Co-variation between elevation and cumulative years of detected WSBW activity accounted for an additional 1% of variation.

### 2.3. Understory Composition, Density and Health

We found corkbark fir, white fir, Engelmann spruce, and Douglas-fir regenerating in the understory (Table 4). Non-host regeneration included quaking aspen and limber pine at higher elevations and ponderosa pine, Gambel oak, New Mexican locust, and blue spruce at lower elevations (Table 4). Although we found all species in each stand, regeneration was spatially variable throughout the stands with 32.8% of plots absent of host species regeneration. We found a similar pattern for quaking aspen with 69.4% of plots absent of regeneration. A mean of 59.5% of all recorded quaking aspen regeneration showed evidence of ungulate browse, insects or pathogens. Defoliation by WSBW was similar between seedlings (mean 14.6%) and saplings (mean 15.0%) across host species in all stands. However, when forest type was considered separately, saplings were more defoliated than seedlings in higher elevation spruce-fir stands (*p* < 0.05) while no difference was found between size classes in mixed-conifer stands. A multivariate model of seedling density explained 19% of variation. Elevation explained 4% of variation while canopy cover explained 2% and cumulative defoliation explained 1% (*p* < 0.05). Co-variation between canopy cover and elevation accounted for an additional 9% of variation. Multivariate models of sapling density explained 13% of variation with elevation and canopy cover accounting for 4% and 2%, respectively (*p* < 0.05). Co-variation between the two explanatory variables accounted for 5% of variation.

## 3. Discussion

### 3.1. Species Composition Shifts by Forest Type

Our results demonstrate that prolonged and repeated outbreaks of WSBW in southwestern forests are capable of changing species composition through widespread mortality; however, the net effect of prolonged disturbance varies by forest type. In spruce-fir stands, we found reductions in density and shifts in species composition towards Douglas-fir and quaking aspen, effectively moving stands away from the spruce-fir forest type. We also found that this shift varies along an elevational gradient. Few live trees remained in the highest elevation stands. In mixed-conifer forests, we showed that outbreaks of WSBW may shift species composition toward dominance of both Douglas-fir and non-host species. Host species continued to regenerate in all stands, despite defoliation and a history of chronic WSBW outbreaks. However, our results suggest that mixed-conifer forests are more resilient to defoliation than adjacent spruce-fir stands and represent a diverging relationship between prolonged outbreaks of WSBW and stand resilience. 

Host susceptibility to WSBW generally occurs on a gradient of shade tolerance (higher defoliation in more shade tolerant species [8,29]) and our results concur, with greater mortality in the more shade tolerant corkbark fir at higher elevations and white fir at lower elevations. Mortality of shade tolerant species has shifted species composition towards less shade-tolerant host or non-host species. At the lower end of the elevation gradient, white fir mortality has increased the importance of Douglas-fir and ponderosa pine, which are also more drought tolerant than white fir. At higher elevations, corkbark fir mortality has resulted in a shift towards Engelmann spruce.

### 3.2. Trends by Elevation

A confounding factor in our study in the Southwest is that as elevation increases, species composition shifts towards a higher percentage of host species, a clear trend in our multivariate models of live and reconstructed species importance values. This effect may be further confounded by the pseudoreplicated analysis of overstory species data. In mixed-conifer, white fir was more common and occupied more basal area in the two lowest elevation stands, whereas Douglas-fir was higher in both metrics in the two highest elevation stands. Ponderosa pine appeared to fall along an elevational gradient (15% reduction from low to high), where wetter and cooler sites at higher elevations limited the distribution. In spruce-fir forest types, the two lowest elevation stands were historically comprised of a higher percentage of Douglas-fir relative to corkbark fir and Engelmann spruce. It is unclear whether our reconstructed importance values represent typical species composition dominance in spruce-fir, as the type has not been studied extensively in the Southwest. In a 1959/1960 survey in the Sangre de Cristo Mountains of northern New Mexico, corkbark fir accounted for 36 and 13% of live basal area on north and south slopes, respectively [30]. The sites examined were located at 3397 m elevation and made no note of WSBW. In comparison, corkbark fir accounted for 20% of live basal area in our Bunker Hill stand (3288 m elevation, NE aspect) and 7% of live basal area in the Mount Taylor stand (3115 m elevation, N aspect). We found standing dead trees to be a major component of all spruce-fir stands. Cumulative years of detected defoliation explained only a small amount of variation in spruce-fir mortality across our gradient, indicating that defoliation, even at relatively low levels, can cause high levels of mortality in spruce-fir stands over time. The spruce-fir forest type is reported to be declining across the southwestern US due to competition and the changing climate [24]. Declines of subalpine fir have also been reported in southwestern Colorado and linked with climatic mismatches and biotic mortality agents; notably, these findings did not include WSBW [31,32]. 

Curiously, we observed that defoliation metrics decreased with increasing elevation in mixed-conifer stands while the lowest elevation spruce-fir stands had the highest defoliation. Defoliation by WSBW has been found to decrease with increasing elevation in central Idaho, north-central Washington and western Montana ([33] as cited in [8,34]). Decreasing defoliation in higher elevation spruce-fir stands likely corresponds to climatic conditions being unfavorable to support large populations of WSBW. The increase in defoliation with decreasing elevation in mixed-conifer stands likely highlights the variability that exists in the interaction between site and WSBW population dynamics. Our mixed-conifer stands only represent a portion of this variability; further research is needed to more fully understand these interactions.

### 3.3. Other Disturbances and Interactions with WSBW

Western spruce budworm is not the sole driver of disturbance or stand dynamics in these stands. Drought, lack of fire, other insects and pathogens, and climate change are all likely interacting factors causing a complex of tree decline, which has been more severe in the last two decades [35]. Although it is hypothesized and reported that defoliation by WSBW increases susceptibility to other insects and pathogens [10,11], we found no indication that bark beetle activity increased in these stands as a result of WSBW. The four primary phloem-feeding beetles of host trees in spruce-fir and mixed-conifer forest types are the spruce beetle (*Dendroctonus rufipennis* Kirby), the western balsam bark beetle (*Dryocoetes confusus* Swaine), the fir engraver (*Scolytus ventralis* (LeConte)) and the Douglas-fir beetle, and these species prefer larger diameter trees [36,37,38,39]. Although the availability of large trees was limited, mortality in our stands was primarily in the lower diameter classes. This lower diameter mortality is more likely to be driven by WSBW and interactions with other biotic agents (sensu [9,40]). The impact of defoliation by WSBW is typically greatest in lower diameter classes [8,9]; however, we found no trend between overstory tree diameter and defoliation, which was unexpected. There are several potential explanations for this result. It is possible that previous tree mortality removed a subset of diameters, obscuring the relationship, or the range of error present in defoliation measurements exceeds the resolution needed to detect mortality differences across diameter size classes. It is also possible that another causal agent was responsible for the small tree mortality, whose indicators were not visible when we collected data.

### 3.4. Understory Condition and Trajectory

We observed low densities of host species regenerating in the understory of most stands. The low density of sapling-sized trees coupled with higher levels of defoliation in the spruce-fir stands may indicate mortality due to WSBW in the sapling size class, likely due to the increased crown area over seedlings. Lower canopy classes, and thus diameter classes, are generally more susceptible to defoliation by WSBW as larvae disperse and fall to lower-strata trees [8]. Thus, small diameter trees experience higher levels of defoliation. Shade-tolerant species, such as corkbark fir or white fir, can persist in the understory for many years. As small trees increase in height, defoliation may prevent recruitment into the overstory, regardless of seedling density. In the absence of shade-tolerant species, quaking aspen may recruit in large areas of mortality as has been documented following prolonged outbreaks of eastern spruce budworm (*Choristoneura fumiferana* (Clem.) in boreal mixedwood forests [41]. However, this does not appear to be the case in our stands, given the high levels of browse and/or insect or pathogen damage observed. The current recruitment levels of host and non-host species in our stands may be insufficient to replace the current mortality of overstory species in the spruce-fir forest type. In these stands, the combined overstory mortality and understory recruitment levels suggest WSBW defoliation is capable of changing stand composition and that these stands will not persist as spruce-fir dominated stands. In mixed-conifer forests, the persistence of host species in these stands indicates that a major compositional shift has not yet occurred.

### 3.5. Impact of WSBW on Resistance and Resilience

The predicted warmer and more arid climate future of the US Southwest [42,43,44] may shift the relationships between host species and WSBW. Since these data were collected, WSBW outbreaks in New Mexico forests have been declining in the affected area [45]; however, it is unlikely that New Mexico forests have seen the end of prolonged or severe WSBW outbreaks. Host susceptibility to WSBW defoliation along the gradient of shade tolerance is inversely correlated with drought tolerance [8,46,47,48]. Defoliation by WSBW may accelerate tree drought stress and mortality in shade-tolerant species, thus increasing the rate of transition to shade intolerant species. Such conditions may lead to increased resilience to fire and drought in mixed-conifer forests (e.g., increased dominance by ponderosa pine, a more fire- and drought-tolerant species). Prolonged and more severe outbreaks may also reduce host tree density to levels that may negatively affect WSBW populations. Interactions among WSBW, other abiotic and biotic disturbances (e.g., spruce beetle, fire), climate and host species’ adaptive capacity have not been widely studied, particularly in the US Southwest. Spruce-fir stands in the Southwest may be increasingly limited by climate, as evidenced by ongoing mortality events linked with climate and regeneration challenges [31,49,50]. Heavily defoliated spruce-fir stands are unlikely to return to a healthy spruce-fir condition, indicating a lack of both resistance and resilience. Although our results are compelling, implications and conclusions are limited by our small sample size. Additional empirical research is needed to more clearly understand the relationships between WSBW, climate, host species, and ultimately, the impact of WSBW on long-term resistance and resilience of these forests under the rapidly changing climate.

## 4. Materials and Methods

### 4.1. Stand Selection

We sampled nine stands across an elevational gradient in west-central and northern New Mexico (Figure 4); forest types were classified as either mixed-conifer or spruce-fir depending upon species composition (Table 1; [24]). All stands were located on US Department of Agriculture Forest Service lands except for Long Canyon, which was located on private land (Vermejo Park Ranch, north-central New Mexico). Stand selection and sampling were completed during the summers of 2012 and 2013. Stands were also selected along a defoliation gradient using US Forest Service Forest Health Protection (USFS FHP) aerial survey data from 1985 to 2011 (referred to as detected defoliation above; USDA Forest Service, Forest Health Protection and New Mexico State Forestry 1985 to 2011 (Available from: https://www.fs.usda.gov/foresthealth/applied-sciences/mapping-reporting/detection-surveys.shtml#idsdownloads (accessed on 25 October 2022). Stands were chosen with the assumption that a higher number of years of WSBW detection would have increased impacts on stand structure, composition and development. The preferred host species of WSBW in the southwestern US include shade-tolerant, fire-intolerant white fir and corkbark fir (*Abies concolor* (Gord. & Glend.) Lindl. ex Hildebr. and *A. lasiocarpa* (Hook.) Nutt. var. *arizonica* (Merriam) Lemmon), Douglas-fir (*Pseudotsuga menziesii* (Mirb.) Franco var. *glauca* (Beissn.) Franco), and Engelmann spruce (*Picea engelmannii* Parry ex Engelm.) [29]. Selected stands also met the following criteria: (1) at least 50% pre-2000 species composition comprised of WSBW host species; (2) no forest management activities within previous 20 years; and (3) similar slope, aspect, and vegetation association [51]. Criteria for species composition, treatments and vegetation association were assessed in the field, while slope, and aspect were identified prior to field visits. We were unable to find areas that had little to no detected defoliation between 1985 and 2011 that also met our stand selection criteria.

Stand elevations ranged from 2629 to 3288 m. Species composition in spruce-fir stands included corkbark fir, Engelmann spruce, Douglas-fir and quaking aspen (*Populus tremuloides* Michx.). Minor components of white fir, Rocky Mountain bristlecone pine (*Pinus aristata* Engelm.) and white pine (*Pinus flexilis* James, *Pinus strobiformis* Engelm or a hybrid of these; [52]) were present in some spruce-fir stands. Species composition in the mixed-conifer stands included ponderosa pine, white fir, Douglas-fir, and quaking aspen. Minor stand components in the overstory included corkbark fir, Rocky Mountain juniper (*Juniperus scopulorum*, Sarg.), Engelmann spruce, and white pine. Gambel oak (*Quercus gambelii* Nutt.), New Mexico locust (*Robinia neomexicana* A. Gray), and blue spruce (*Picea pungens* Engelm.) were found in the understory.

Spruce-fir stands were primarily north-facing except Long Canyon, which faced east (Table 1). Slopes ranged from 16 to 20% on average. Soils in Mount Taylor and Cold Spring were comprised of vitrandic eutrocryepts and haplocryolls [53]. Soils in Bobcat Canyon were complex and comprised of cobbly sandy loam with neutral acidity, and soils in Bunker Hill ranged from very cobbly sandy loam with neutral acidity to cobbly loam with very strong acidity [53]. Long Canyon soils ranged from gravelly sandy loam with neutral acidity to cobbly loam with very strong acidity [53]. Interpolated winter precipitation ranged from 142.7 to 187.4 mm, and mean maximum temperature ranged from 19.8 to 21.8 °C [54].

Aspect in mixed-conifer stands ranged generally from east to southeast (90⁰ to 160⁰), with slopes ranging from 15 to 20% on average (Table 1). Cimarron soils were comprised of stony sandy loam. Jose Maria soils were well-drained, very cobbly loam to very gravelly clay loam. Valle Grande and Vallecitos were both comprised of well drained sandy loam. Interpolated winter precipitation ranged from 131.7 to 189.8 mm and mean maximum temperature ranged from 22.4 to 24.9 °C [54].

### 4.2. Field Methods

A two-stage cluster sampling design was implemented in each stand, as this design was previously found to increase sampling efficiency of forest composition in WSBW-affected stands [55]. Using stand boundaries, a systematic random grid of ten, 0.02 ha plots were installed in each stand (“grid plots”). Grid points were generated using the Hawth’s tools 3.27 [56] and ArcGIS 10.0 [57]. A second 0.02 ha plot was installed 50 m away in a random azimuth from each grid plot (“cluster plots”). Five nested 0.001 ha regeneration plots were installed in each grid and cluster plot. Regeneration plots were located at plot center and 4 m from plot center in each cardinal direction. Due to extreme slope, four plots in Jose Maria were not sampled and two plots in Cold Spring were not sampled.

For each 0.02 ha plot, we recorded abiotic characteristics including slope, aspect, elevation, and GPS coordinates. Vegetation association was recorded [51], and canopy cover was measured using a GRS densitometer (Geographic Resource Solutions GRS, Arcata, CA, USA). Densitometer measurements were taken at 1 m spacing on two 16 m transects bisecting plot center north to south and east to west, excluding a measurement at plot center on the east to west transect to avoid confounding measurements, for a total of 29 densitometer points per plot. We recorded status (live or dead), diameter at breast height in centimeters (1.37 m, DBH), tree height in meters, and height to live crown in meters for each tree greater than 12.7 cm DBH. Standing dead trees were rated using a decay class (2 to 5 scale) to estimate the approximate time of mortality [58]. We assessed snags for any insect or pathogen damage by removing bark and visually identifying galleries and/or other evidence of insects or pathogens. Living trees were assessed for insect damage or pathogen damage when symptoms or signs were present [59]. Defoliation of live trees was classified into two classes: current and cumulative. Current defoliation was defined as defoliation identifiable by red foliage and the appearance of frass in the live crown. Cumulative defoliation included the current defoliation in addition to apparent defoliation along multiple years which included dead tops and branches. Defoliation of live trees by WSBW was measured using two methods. The first method, developed to reduce bias present in visual dieback estimates, was adapted from techniques outlined by [60] (sketch method) and completed in all stands. Individual tree cumulative defoliation was sketched on a transparency using the guidelines of [61] and later digitally scanned at 300 dots per inch. Pixels were counted using the analysis tools in Adobe® Photoshop CS5. The proportion of defoliated crown area to total crown area was recorded and used to estimate percent canopy defoliation. Although the measurements are presented as a continuous variable, the method only reduces bias between individual assessments and is still prone to bias [62]. The second method incorporated visual estimates of current and cumulative defoliation in 25% categories (0%, 1–25%, 26–50%, 51–75%, 76–99%, 100%). Categories were estimated for each crown third and then averaged for the total tree (6-class method) [63]. We estimated defoliation using the 6-class method in all stands except for Vallecitos. Sketch method defoliation and 6-class method were evaluated for similarity using a Kendall’s rank-based correlation.

In each 0.001 ha plot, we recorded height of seedlings (trees < 1.37 m) by 0.15 m height class and DBH (cm) of each sapling (trees > 1.37 m and < 12.7 cm DBH). We visually estimated individual defoliation for host seedlings and saplings in 10% categories. Non-host seedlings were tallied by height class and saplings were measured as individuals. Both seedlings and saplings of quaking aspen were rated as healthy or declining. The declining category included evidence of ungulate browse (missing leaders, chewed or multi-branched stems), insects (evidence of herbivory or insect sign) or pathogens (yellowed or chlorotic leaves or other signs of pathogens) [59]. Regeneration of host species was measured on both grid and cluster plots, and regeneration of non-host species was measured on grid plots only.

Original data are publicly available through the Environmental Data Initiative [64].

### 4.3. Data Analysis

We treated grid plots and cluster plots as independent rather than as part of a cluster, due to the spacing of our systematic random grid within a stand, which ranged from 140 to 250 m between grid plots. The combination of the small spacing between grid plots and the 50 m distance between grid and cluster plots could result in a cluster plot randomly lying between grid points, thus losing the hierarchical nature of the design [65].

To understand the species composition of our overstory plots, we utilized modified importance values (IV) developed by [66] for each species, where:IV = relative dominance (R_dom_) + relative density (R_den_)(1)
R_dom_ = (basal area (m^2^ha^−1^) per species/total basal area (m^2^ha^−1^)) ∗ 100(2)
R_den_ = (trees ha^−1^ per species/total trees ha^−1^) ∗ 100(3)

To calculate importance values at the plot level and measure variability within a stand, we omitted relative frequency from the original importance value calculations. Therefore, a species measured as the only species within a plot would have a maximum importance value of 200. Importance values were calculated for live and reconstructed (live and standing dead) host species to understand changes in species composition as a result of disturbance. We report the difference between live and reconstructed importance values. We transformed aspect [67] for each plot at 180° to account for the higher severity of defoliation by WSBW on dry, south-facing sites [8]. Aspect transformation allows for a sigmoidal representation of aspect where 0 is the opposing aspect (0° in the case of 180° transformed aspect) and 2 at the target transformation (180° in the case of 180° transformed aspect) [68]. We also transformed aspect at 45° to account for the increase in forest growth generally found on north and east slopes. Stand-level canopy cover was calculated by first calculating plot-level canopy cover using the percentage of canopy cover detected using the densitometer for each transect and averaging the two transects, then averaging across plots.

Much of our data violated the normality assumptions of traditional ANOVA and regression. In order to understand similarities and differences in abiotic and biotic conditions among stands within a forest type, we compared abiotic stand characteristics (slope, aspect, elevation), host species composition, mean canopy cover, and mean years detected defoliation using Wilcoxon rank-sum tests. Values between species and tree sizes (understory and overstory) across all stands were compared using Wilcoxon rank-sum tests and correlations assessed between variables using Kendall’s rank-based correlation. All Wilcoxon rank-sum tests are shown with Bonferroni-corrected *p*-values [64]. All statistical analyses were performed in the R statistical computing environment, version 4.0.2 [67]. We modelled live importance values, reconstructed importance values, seedling abundance and sapling abundance for the four most abundant species within forest types (spruce-fir: Engelmann spruce, Douglas-fir, corkbark fir and quaking aspen; mixed-conifer: white fir, Douglas-fir, ponderosa pine and quaking aspen) using distance-based redundancy analysis using the R package *vegan* [69]. Bray–Curtis measures of dissimilarity were used to calculate dissimilarity matrices for each set of dependent variables. A value of 0.01 was added to each dependent variable to account for plots that contained zero trees [70]. We relativized our environmental data by column total (between 0 and 1) due to the differences in measurement scales. Cumulative years of detected WSBW defoliation, sketch method defoliation, elevation, 0° transformed aspect, and slope were examined as explanatory variables. Forward-selection procedures were used to aid in model fitting and model selection was based on the Akaike Information Criterion (AIC; [71]). The effect of each independent variable in each of the final models was estimated using canonical variance partitioning in the R package *vegan* [68].

## 5. Conclusions

Our results suggest that prolonged or repeated outbreaks of WSBW may accelerate transitions from shade-tolerant to a more shade intolerant species composition, resulting in more drought and WSBW resistant forest structure in lower elevation mixed-conifer forests. In higher elevation spruce-fir forests, repeated or prolonged WSBW outbreaks may result in an elimination or severe reduction of shade-tolerant species from the landscape and ultimately forest type conversions away from spruce-fir.

## Figures and Tables

**Figure 1 plants-11-03266-f001:**
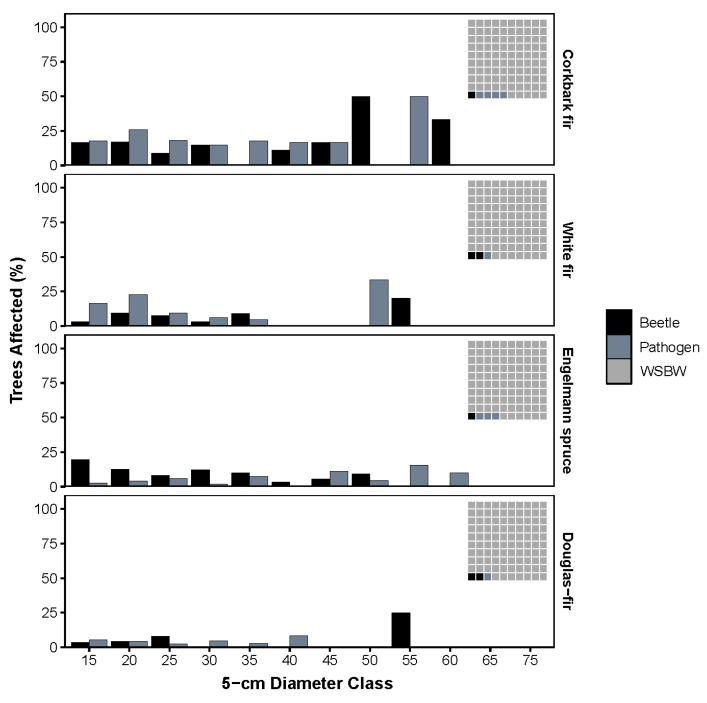
Percent of all trees (live and dead) by species and 5 cm diameter class, affected by bark beetles and pathogens. ***Inset waffle plots*** display proportion of all live trees affected by western spruce budworm (WSBW; light gray), bark beetles (black), or pathogens (dark grey). WSBW defoliation by size class not represented to aid in *y*-axis interpretation.

**Figure 2 plants-11-03266-f002:**
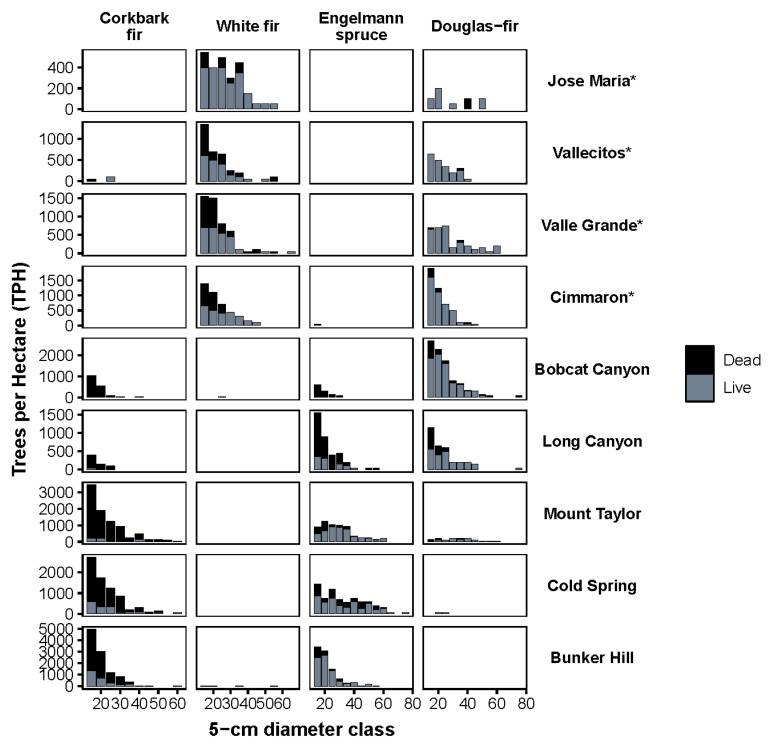
Diameter distributions of overstory (>12.7 cm DBH) live (grey bars) and standing dead (black bars) trees ha^−1^ by 5 cm diameter. * denotes mixed-conifer stands. Note differences in *y*-axis scales.

**Figure 3 plants-11-03266-f003:**
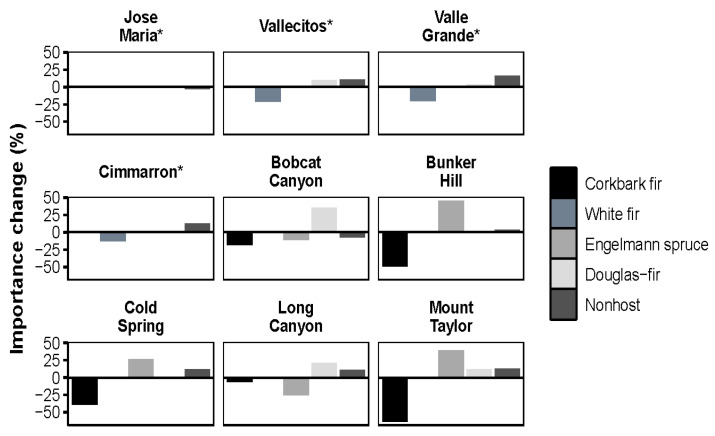
Percent change (difference) in relative importance value of overstory by species. Negative values reflect declines in importance while positive values reflect increases in importance. * indicates mixed-conifer stands.

**Figure 4 plants-11-03266-f004:**
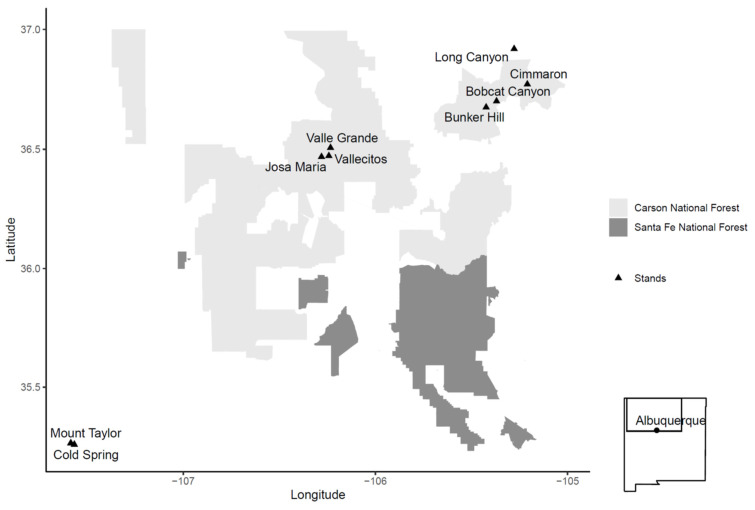
Locations of nine stands with western spruce budworm (*Choristoneura freemani)* outbreaks sampled in northern New Mexico, USA.

**Table 1 plants-11-03266-t001:** Stand level metrics (mean and standard error) for nine stands sampled in northern New Mexico, USA. Different letters indicate significant differences between stands and within forest types using Wilcoxon rank-sum test (α < 0.05).

Stand	Forest Type	Mean Duration Detected Defoliation (Years)	Elevation (m)	Host Species Composition (%)	Canopy Cover (%)	Aspect (⁰)	Slope (%)
Jose Maria	Mixed-conifer	14.3 _a_	2629 (1.8) _a_	78.5 (0.1) _ab_	51.7 (1.6) _a_	91	14.4 (0.5) _a_
Vallecitos	Mixed-conifer	9.7 _bcde_	2792 (0.8) _b_	59.3 (0.1) _a_	33.4 (1.0) _a_	160	28.0 (0.5) _b_
Valle Grande	Mixed-conifer	8.6 _b_	2817 (3.0) _b_	65.5 (0.1) _a_	53.6 (0.7) _a_	102	18.8 (0.3) _ab_
Cimarron	Mixed-conifer	9.5 _bc_	2921 (2.0) _c_	82.3 (0.1) _ab_	37.6 (1.1) _a_	100	17.1 (0.4) _ab_
Bobcat Canyon	Spruce-fir	12.2 _ad_	3017 (4.8) _d_	86.3 (0.0) _ab_	51.5 (3.9) _bc_	17	17.4 (1.5) _ab_
Long Canyon	Spruce-fir	7.1 _e_	3053 (3.8) _e_	55.5 (0.1) _a_	34.6 (3.7) _b_	16	15.6 (2.6) _ab_
Mount Taylor	Spruce-fir	11.6 _acd_	3115 (13.1) _ef_	80.1 (0.1) _ab_	57.6 (3.4) _c_	18	19.8 (2.3) _ab_
Cold Spring	Spruce-fir	9.7 _c_	3173 (11.1) _f_	83.7 (0.1) _ab_	51.4 (3.2) _bc_	20	16.3 (1.7) _a_
Bunker Hill	Spruce-fir	8.7 _bce_	3288 (9.9) _g_	92.9 (0.0) _b_	51.2 (3.5) _bc_	38	20.1 (1.8) _ab_

**Table 2 plants-11-03266-t002:** Mean cumulative defoliation [%; (standard error)] by stand, species, and defoliation assessment method (Sketch and 6-class) for 9 stands in northern New Mexico, USA. See text for details on defoliation assessments.

	Corkbark Fir		White Fir		Engelmann Spruce		Douglas-Fir	
Stand	Sketch	6-Class	Sketch	6-Class	Sketch	6-Class	Sketch	6-Class
Jose Maria	–	–	19.3 (1.9)	26–50%	–	–	11.7 (9.6)	26–50%
Vallecitos	15.5 (3.3)	–	37.1 (3.2)	–	–	–	23.6 (2.9)	–
Valle Grande	–	–	15.6 (1.7)	26–50%	–	–	7.4 (1)	1–25%
Cimarron	–	–	9.5 (2.2)	1–25%	–	–	7.7 (1.1)	1–25%
Bobcat Canyon	–	–	–	–	28.7 (–)	26–50%	22.1 (0.8)	26–50%
Long Canyon	16.7 (–)	26–50%	–	–	6.8 (1.5)	26–50%	12.1 (1)	26–50%
Mount Taylor	35.2 (5.2)	51–75%	–	–	35.2 (2.2)	26–50%	14.9 (2.6)	26–50%
Cold Spring	13.2 (1.2)	26–50%	–	–	11.8 (0.7)	26–50%	14 (6.5)	26–50%
Bunker Hill	12.2 (1)	26–50%	8.3 (2.1)	26–50%	10.9 (0.6)	26–50%	–	–

**Table 3 plants-11-03266-t003:** Overstory (>12.7 cm diameter at breast height) live and standing dead tree density per hectare (TPH) and mean basal area (BA, m^2^ ha^−1^) (standard error) for a. four mixed-conifer stands and b. five spruce-fir stands sampled in northern New Mexico, USA. Stands are ordered from low to high elevation starting with Jose Maria (lowest) and ending with Bunker Hill (highest).

**a. Mixed-Conifer**	**Jose Maria**		**Vallecitos**		**Valle Grande**		**Cimarron**			
	TPH	BA	TPH	BA	TPH	BA	TPH	BA		
** *White fir* **										
Live	143.3 (32.7)	9.2 (2.3)	92.5 (22.4)	4.1 (1.0)	130.0 (18.6)	6.4 (1.4)	127.5 (32.3)	7.1 (1.9)		
Dead	30.0 (11.8)	1.4 (0.8)	75.0 (17.2)	3.6 (1.1)	115.0 (23.0)	4.8 (1.4)	82.5 (36.1)	2.3 (1.2)		
** *Douglas-fir* **										
Live	30.0 (16.0)	2.1 (1.4)	100.0 (26.9)	4.3 (1.2)	162.5 (32.4)	12.1 (2.1)	205.0 (44.1)	8.0 (1.7)		
Dead	6.7 (4.5)	0.8 (0.5)	2.5 (2.5)	0.2 (0.2)	5.0 (3.4)	0.3 (0.2)	25.0 (11.2)	0.8 (0.5)		
** *Ponderosa pine* **										
Live	50.0 (23.4)	5.0 (2.4)	105.0 (22.3)	10.1 (2.6)	27.5 (9.2)	2.4 (0.9)	15.0 (8.2)	0.8 (0.4)		
Dead	6.7 (4.5)	1.0 (0.8)	7.5 (4.1)	0.5 (0.3)	2.5 (2.5)	0.2 (0.2)	0 (0)	0 (0)		
** *Quaking aspen* **										
Live	16.7 (13.5)	1.0 (0.9)	27.5 (17.6)	0.9 (0.5)	122.5 (25.0)	6.7 (1.6)	20.0 (12.2)	1.6 (1.0)		
Dead	0 (0)	0 (0)	25.0 (12.3)	0.6 (0.3)	40.0 (16.1)	1.4 (0.6)	10.0 (5.8)	0.6 (0.4)		
** *Other species ** **										
Live	3.3 (3.3)	0.1 (0.1)	5.0 (5.0)	0.3 (0.3)	0 (0)	0 (0)	60.0 (26.1)	2.0 (0.9)		
Dead	0 (0)	0 (0)	2.5 (2.5)	0 (0)	0 (0)	0 (0)	2.5 (2.5)	0 (0)		
**b. Spruce-fir**	**Bobcat Canyon**		**Long Canyon**		**Mount Taylor**		**Cold Spring**		**Bunker Hill**	
	TPH	BA	TPH	BA	TPH	BA	TPH	BA	TPH	BA
** *Corkbark fir* **										
Live	0 (0)	0 (0)	2.5 (2.5)	0 (0)	35.0 (10.3)	2.3 (0.8)	91.7 (28.9)	5.1 (1.4)	137.5 (41.2)	4.8 (1.7)
Dead	90.0 (38.0)	2.5 (0.9)	30.0 (13.3)	0.8 (0.4)	402.5 (48.7)	19.1 (2.7)	319.4 (51.8)	14.0 (2.2)	395.0 (68.4)	13.5 (2.4)
** *Engelmann spruce* **										
Live	2.5 (2.5)	0.2 (0.2)	47.5 (12.8)	2.0 (0.5)	235.0 (41.5)	20.3 (3.2)	263.9 (62.1)	27.4 (4.3)	395.0 (50.4)	17.4 (2.3)
Dead	55.0 (14.5)	1.7 (0.5)	135.0 (26.7)	5.5 (1.2)	82.5 (29.3)	4.5 (1.8)	147.2 (33.7)	13.2 (4.0)	87.5 (13.5)	2.8 (0.6)
** *Douglas-fir* **										
Live	380.0 (39.6)	21.1 (2.4)	112.5 (30.5)	7.5 (2.4)	52.5 (21.6)	5.7 (1.9)	5.6 (3.8)	0.2 (0.2)	0 (0)	0 (0)
Dead	80.0 (18.3	4.5 (1.4	47.5 (15.6)	1.2 (0.4)	12.5 (6.2)	0.5 (0.3)	0 (0)	0 (0)	0 (0)	0 (0)
** *Quaking aspen* **										
Live	87.5 (31.2)	2.5 (1.0)	177.5 (41.6)	8.3 (1.9)	127.5 (35.6)	6.5 (1.9)	75.0 (24.3)	5.5 (1.8)	40.0 (24.0)	0.8 (0.5)
Dead	100.0 (27.9)	2.1 (0.6)	152.5 (31.7)	4.3 (0.9)	72.5 (21.3)	2.2 (0.7)	19.4 (10.0)	0.5 (0.3)	17.5 (10.4)	0.4 (0.3)
** *Other species ** **										
Live	20.0 (11.1)	1.1 (0.8)	20.0 (11.1)	0.6 (0.4)	0 (0)	0 (0)	0 (0)	0 (0)	20.0 (14.2)	1.3 (0.9)
Dead	7.5 (5.5)	0.6 (0.5)	0 (0)	0 (0)	0 (0)	0 (0)	0 (0)	0 (0)	7.5 (5.5)	0.4 (0.3)

* Other species include Rocky Mountain bristlecone pine in Long Canyon, Rocky Mountain bristlecone pine, white pine (limber pine, southwestern white pine or a hybrid of these) and white fir in Bunker Hill, and white pine (limber pine, southwestern white pine or a hybrid of these), white fir and an unknown juniper species in Bobcat Canyon.

**Table 4 plants-11-03266-t004:** Mean understory live seedling and sapling tree density per hectare (standard error) for a. four mixed-conifer stands and b. five spruce-fir stands sampled in northern New Mexico, USA. Seedlings = trees < 1.37 m tall; saplings = trees > 1.37 m tall and <12.7 cm diameter at breast height. Stands are ordered from low to high elevation starting with Jose Maria (lowest) and ending with Bunker Hill (highest).

**a. Mixed-Conifer**	**Jose Maria**	**Vallecitos**	**Valle Grande**	**Cimarron**	
** *White fir* **					
Sapling	330.0 (79.2)	100.0 (38.9)	30.0 (17.1)	120.0 (49.8)	
Seedling	1040.0 (150.4)	210.0 (71.5)	50.0 ( 21.9)	250.0 (70.2)	
** *Douglas-fir* **					
Sapling	110.0 (49.0)	90.0 (37.9)	0.0 (0.0)	60.0 (34.3)	
Seedling	180.0 ( 47.9)	40.0 (19.7)	50.0 (21.9)	100.0 (30.2)	
** *Ponderosa pine* **					
Sapling	10.0 (10.0)	10.0 (10.0)	0.0 (0.0)	10.0 (10.0)	
Seedling	20.0 (14.1)	0.0 (0.0)	0.0 (0.0)	0.0 (0.0)	
** *Quaking aspen* **					
Sapling	20.0 (14.1)	10.0 (10.0)	0.0 (0.0)	10.0 (10.0)	
Seedling	480.0 (103.9)	40.0 (31.5)	880.0 (151.9)	60.0 (31.2)	
** *Other species ** **					
Sapling	50.0 (21.9)	260.0 (86.0)	0.0 (0.0)	0.0 (0.0)	
Seedling	10.0 (10.0)	210.0 (64.0)	0.0 (0.0)	200.0 (72.5)	
**b. Spruce-fir**	**Bobcat Canyon**	**Long Canyon**	**Mount Taylor**	**Cold Spring**	**Bunker Hill**
** *Corkbark fir* **					
Sapling	0.0 (0.0)	20.0 (20.0)	80.0 (38.8)	320.0 (129.2)	921.6 (227.2)
Seedling	1100.0 (227.2)	580.0 (212.1)	2520.0 (604.2)	380.0 (207.8)	11660.0 (2304.3)
** *Engelmann spruce* **					
Sapling	20.0 (20.0)	160.0 (77.5)	180.0 (68.2)	40.0 (28.0)	254.9 (107.9)
Seedling	680.0 (214.6)	700.0 (254.3)	460.0 (171.9)	40.0 (28.0)	1560.0 (390.9)
** *Douglas-fir* **					
Sapling	40.0 (28.0)	0.0 (0.0)	0.0 (0.0)	0.0 (0.0)	0.0 (0.0)
Seedling	1060.0 (204.9)	420.0 (151.4)	220.0 (131.8)	0.0 (0.0)	20.0 (20.0)
** *Quaking aspen* **					
Sapling	0.0 (0.0)	0.0 (0.0)	0.0 (0.0)	0.0 (0.0)	0.0 (0.0)
Seedling	1220.0 (305.4)	2000.0 (344.0)	5140.0 (1386.5)	1280.0 (330.8)	1200.0 (387.6)

* Other species include limber pine and corkbark fir in Cimmaron, Gambel oak, New Mexico locust and blue spruce in Jose Maria, and Gambel oak in Vallecitos.

## Data Availability

The data presented in this study are openly available in The Environmental Data Initiative at https://doi.org/10.6073/pasta/43e3c0a4be762a8bf5d915051ecb2fc5.

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
