# Peer review of "Western Spruce Budworm Effects on Forest Resilience"

_plants, 2022, doi:10.3390/plants11233266_

Round 1
Reviewer 1 Report
There are several opportunities where the authors might cite Nealis & Regniere (2004), who showed in the eSBW system in Canada that budworm mortality of fir and spruce led to a stand conversion toward aspen-dominated mixed conifer. I would suggest doing this in the discussion.
Author Response
Thank you for this suggestion. We have added a citation that supports the following, revised sentence: "In the absence of shade-tolerant species, quaking aspen may recruit in large areas of mortality as has been documented following prolonged outbreaks of eastern spruce budworm (Choristoneura fumiferana (Clem.) in boreal mixedwood forests [41]. However, this does not appear to be the case in our stands, given the high levels of browse, and / or insect or pathogen damage observed." L296-299.
Reviewer 2 Report
In this study, author evaluated the effects of severe western spruce budworm outbreaks on stand dynamics in the US Southwest. Results suggest that prolonged or repeated outbreaks of WSBW may accelerate transitions from shade-tolerant to a more shade intolerant species composition, resulting in more drought and WSBW resistant forest structure in lower elevation mixed-conifer forests. In higher elevation spruce-fir forests, repeated or prolonged WSBW outbreaks may result in an elimination or severe reduction of shade-tolerant species from the landscape and ultimately foresttype conversions away from spruce-fir. But the article is not well organized. experimental methods lack references, experimental methods lack references. Finally, there are some essential problems should be addressed by authors, which are listed below.
1. The sentences in Abstract are not coherent enough, so it is suggested to be revised.
2. It is recommended to add line numbers to facilitate review.
3. P1 “The most widespread native defoliator of forests in western North America is the western spruce budworm” This sentence is abrupt.
4. The sentences in Introduction are not coherent enough, so it is suggested to be revised.
5. P9 “653. Discussion” Errors of writing.
6. Figure 4. is a little blurry.
Round 2
Reviewer 2 Report
Nice work.